# Household cooking fuel estimates at global and country level for 1990 to 2030

Oliver Stoner [1,2 ✉], Jessica Lewis[3], Itzel Lucio Martínez[3], Sophie Gumy[3], Theo Economou[4,2] & Heather Adair-Rohani[3]

Household air pollution generated from the use of polluting cooking fuels and technologies is a major source of disease and environmental degradation in low- and middle-income countries. Using a novel modelling approach, we provide detailed global, regional and country estimates of the percentages and populations mainly using 6 fuel categories (electricity, gaseous fuels, kerosene, biomass, charcoal, coal) and overall polluting/clean fuel use – from 1990-2020 and with urban/rural disaggregation. Here we show that 53% of the global population mainly used polluting cooking fuels in 1990, dropping to 36% in 2020. In urban areas, gaseous fuels currently dominate, with a growing reliance on electricity; in rural populations, high levels of biomass use persist alongside increasing use of gaseous fuels. Future projections of observed trends suggest 31% will still mainly use polluting fuels in 2030, including over 1 billion people in Sub-Saharan African by 2025.

[1] School of Mathematics and Statistics, University of Glasgow, Glasgow, UK. [2] Department of Mathematics, University of Exeter, Exeter, UK. [3] Department of Environment, Climate Change and Health, World Health Organization, Geneva, Switzerland. [4] Climate and Atmosphere Research Centre, The Cyprus Institute, Nicosia, Cyprus. ✉email: oliver.stoner@glasgow.ac.uk

For 3 billion people[1] living in low-income and middle-income countries (LMICs), the simple act of cooking is a major health and safety risk. The inefficient combustion of solid fuels (wood, coal, charcoal, dung, and crop waste) and kerosene in simple stoves and devices produces high levels of household air pollution (HAP). Chronic exposure to HAP increases the risk of noncommunicable disease including ischemic heart disease, stroke, chronic obstructive pulmonary disease, lung cancer, as well as pneumonia[2]. Overall, HAP exposure accounts for some 3.8 million premature deaths annually[3]. Use of open fires or poorly balanced pots is also a major cause of burns and scalds in LMICs, while kerosene and charcoal use in the home is a major source of poisonings from either ingestion or carbon monoxide exposure[2].

Households that rely on polluting energy systems frequently have to travel great distances to gather fuel—sometimes traveling hours each week—putting them at increased risk of musculoskeletal injury and violence[4]. Fuel collection is often tasked to women and children, perpetuating the negative socioeconomic and gender inequities of energy poverty by taking away time that could be spent on other activities like schooling, income-generation, and socializing.[4] Polluting cooking practices are also an important cause of environmental degradation and climate change: the black carbon from cooking, heating and lighting is responsible for 25% of anthropogenic global black carbon emissions[5], and around 30% of wood fuels harvested globally are unsustainable[6].

In recognition of these significant burdens, the global community has prioritized achieving universal access to clean cooking, enshrined in the 2030 Agenda for Sustainable Development[7] as one of three targets for Sustainable Development Goal (SDG) 7, to "ensure access to affordable, reliable, sustainable, and modern energy". As part of its mandate to monitor and inform policy towards this goal, the World Health Organization (WHO) publishes estimates of exposure to HAP[8] and related disease burdens[3]. Historically, such estimates were calculated using the estimated population mainly using solid fuels[9] for cooking. However in 2014, the WHO published the first-ever normative guidelines on the fuels and technologies that can be considered "clean" for health[2], which highlight the importance of stove and fuel performance in combination, while also recommending against or discouraging the use of certain fuels—notably unprocessed coal and kerosene, a liquid fuel previously considered clean that emits high levels of harmful pollution. Since then, tracking of "solid fuels" has been replaced with "polluting fuels and technologies"—where polluting fuels consists of unprocessed biomass (wood, crop residues, and dung), charcoal, coal, and kerosene (Fig. 1), and polluting technologies refers to those stoves with emission rates higher than the recommended rates included in WHO Guidelines. Meanwhile, estimates of the proportion of the population mainly using clean fuels and technologies—where clean fuels consists of gaseous fuels (liquified petroleum gas or LPG, natural gas, biogas), electricity, alcohol, and solar energy (Fig. 1)—inform monitoring of progress towards universal access to clean cooking[1]. Acknowledging the very limited survey data on the technologies used for cooking, and the limited availability of truly clean-burning (for health) biomass stoves in LMICs, this analysis focuses only on the fuels used rather than stove technologies.

While the aggregate indicators "polluting fuel use" and "clean fuel use" are effective for summarizing and communicating the global extent of polluting cooking, and progress towards global goals, fuel-specific estimates are needed to optimally inform policies and decision-making on how to achieve the greatest reductions in HAP exposure as quickly as possible. These data in combination with local expert knowledge on challenges of affordability, availability, infrastructure, and cultural preferences are critical to maximizing the health benefits from the clean cooking transition. Fuel-specific estimates are also desirable to refine estimates of HAP exposure and health burdens at regional, country level, and sub-national levels, fully taking into account the varying harm and types of pollution associated with different fuels (notably, carbon monoxide is currently absent from burden of disease calculations[10]).

Using a new model based on individual/specific fuel categories[11] (detailed in "Methods" section), we report estimates of main cooking fuel use at country, regional, (SDG and WHO regions) and global levels, for each year from 1990 to 2020, with urban/rural disaggregation. We provide estimates of aggregate clean and polluting fuel use, and report for the first time estimates for six specific fuel categories: electricity, gaseous fuels, kerosene, unprocessed biomass, charcoal, and coal. For brevity, gaseous fuels and unprocessed biomass are from here onwards called "gas" and "biomass", respectively. We also report future projections of all estimates up to 2030 representing a possible scenario, where trends seen in recent decades continue. We provide all estimates as Supplementary Data for download: Supplementary Data 1 contains estimates at country level; Supplementary Data 2 contains estimates at SDG region and global levels; and Supplementary Data 3 contains estimates at WHO region level.

In this article, we will often refer to percentages or populations *mainly* using different fuels for cooking. This is because the vast majority of existing household surveys, the primary input data for the model, do not capture use of fuels other than the one used most often by the household. Stove-stacking, where a household uses more than one fuel and stove type in parallel with their main

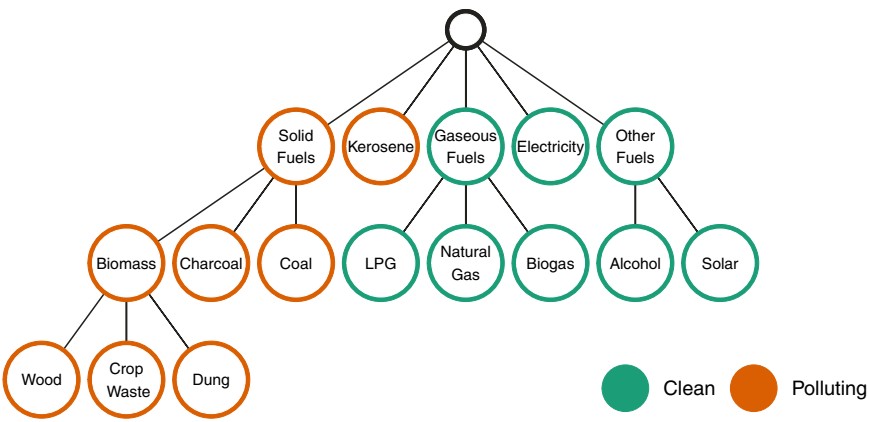

**Fig. 1 Cooking fuel categorization.** Classification of cooking fuels within the scope of the global household energy model as clean or polluting.

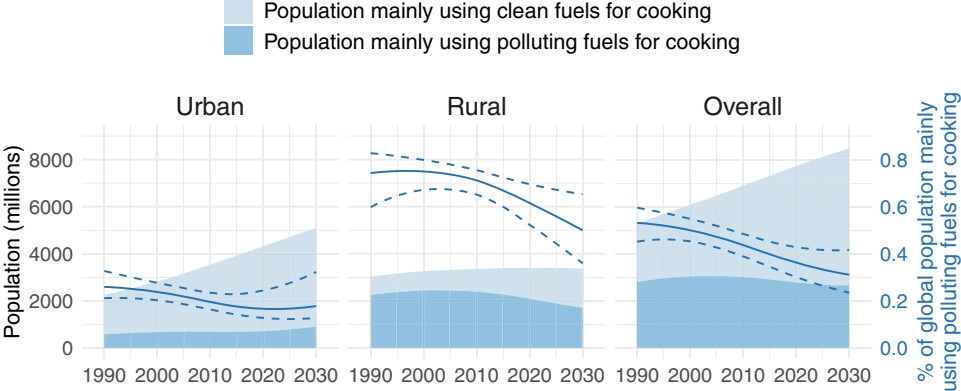

**Fig. 2 Global use of clean and polluting fuels as the main fuel for cooking.** Estimated (posterior median) global populations mainly using clean and polluting fuels for cooking (shaded area), shown alongside the estimated (posterior median) percentage of the global population mainly cooking with polluting fuels (solid line), with 95% posterior uncertainty intervals (dotted lines).

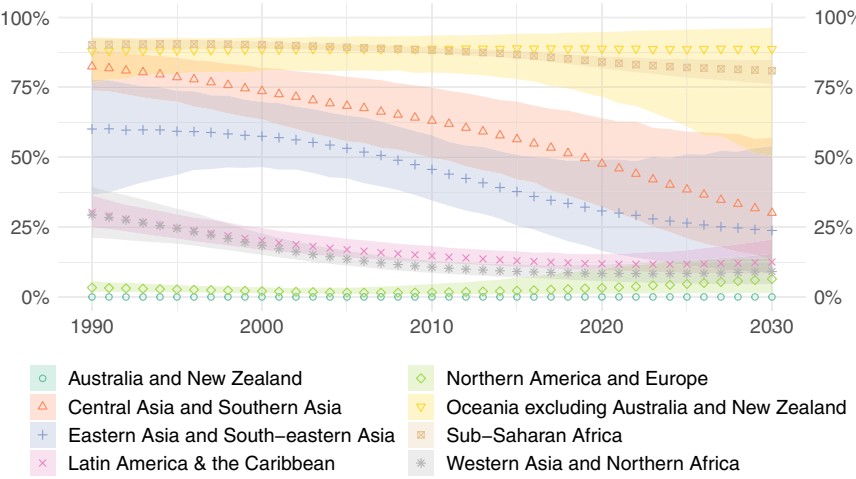

**Fig. 3 Regional use of polluting fuels as the main fuel for cooking.** Estimated (posterior median) percentage of the global population mainly cooking with polluting fuels in each SDG region, with 95% uncertainty intervals (shaded).

fuel, e.g., use of LPG as the main fuel alongside use of a traditional biomass stove[12], is common around the world[13]. Eventually, enough surveys capturing this information will be available to enable comprehensive global estimates (i.e., by country and year) which quantify stove-stacking. Until then, we are limited to quantifying main fuel use and we must recognize that the absolute number of people who use polluting fuels for cooking (and are therefore exposed to high levels of household air pollution) is certainly higher than just the population using them as their main fuel for cooking.

### Results
**Progress towards universal clean fuel use.** The percentage of the global population mainly using polluting fuels for cooking has declined steadily over the last three decades, as illustrated in the right panel of Fig. 2, from 53% [45–60] in 1990 to 36% [30–43] in 2020. If observed trends continue, this percentage is expected to decline further to 31% in 2030. However, the percentage of the population mainly using polluting cooking fuels does not tell the whole story, as rising populations have contributed to an absolute number of people mainly using polluting fuels, which has deviated little from 3 billion people since 1990 (2.8 billion [2.4–3.1] in 1990, 3.0 billion [2.8–3.3] in 2000, 3.0 billion [2.7–3.3] in 2010, and 2.8 billion [2.3–3.3] in 2020). This number is projected to drop only to 2.7 billion people by 2030.

Strictly at a global scale, the percentage of people in rural areas mainly using polluting fuels for cooking (central panel of Fig. 2) decreased only slightly between 1990 and 2010, from 75% [60–83] to 71% [66–76], but progress has since accelerated so that the estimated percentage cooking mainly with polluting fuels in 2020 is 61% [52–69]. This is projected to decrease further to around 50% in 2030. These reductions have been matched by substantial decreases in the absolute rural population mainly using polluting fuels, from a high of 2.5 billion [2.2–2.6] in 2003, to 2.1 billion [1.8–2.4] in 2020 and then a projected 1.7 billion in 2030.

Conversely, following a decrease from 1990 to 2020, the percentage of the global urban population mainly using polluting fuels appears to have plateaued at 17% [13–25] in 2020—projected to be 18% in 2030—while the absolute urban population mainly using polluting fuels is even projected to increase from 0.7 billion [0.5–1.1] in 2020 to 0.9 billion in 2030.

The stagnation in the global population mainly using polluting and clean fuels disguises an important regional trends. In 1990, more than three quarters of people in the Central Asia and Southern Asia region and more than half of people in the Eastern Asia and South-eastern Asia region mainly used polluting fuels for cooking (Fig. 3). Both of these regions have made significant progress over the last three decades in transitioning towards universal use of clean fuels as the main fuel for cooking. However, these successes are overshadowed by alarmingly little progress in

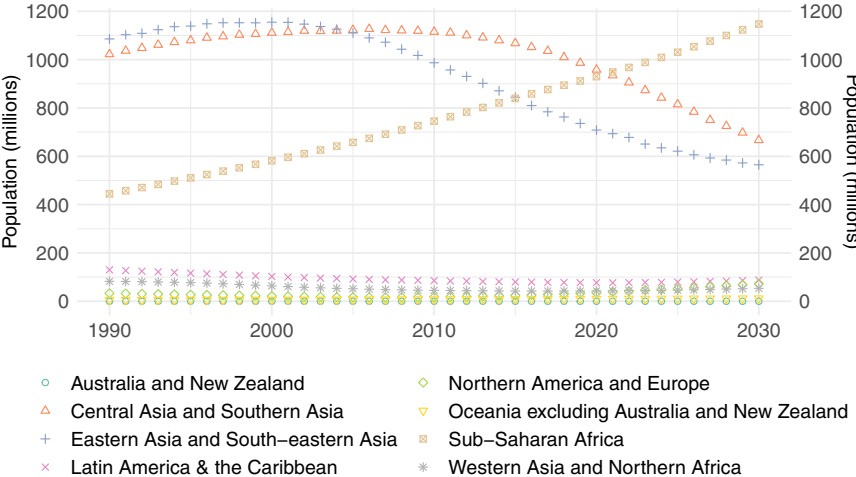

**Fig. 4 Regional populations mainly using polluting fuels for cooking.** Estimated (posterior median) population mainly cooking with polluting fuels in each SDG region.

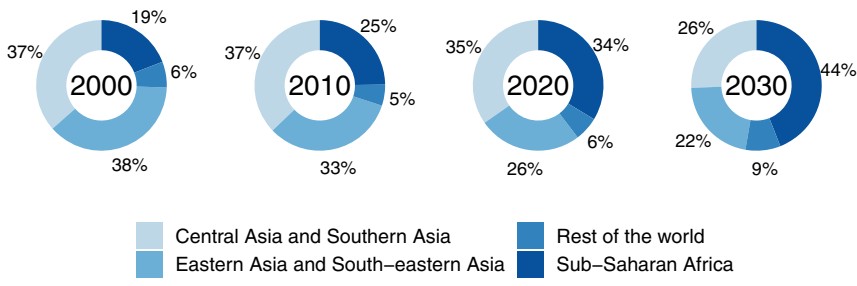

**Fig. 5 Regional breakdown of the global population mainly using polluting fuels for cooking.** Estimated (posterior median) regional populations mainly using polluting fuels as a proportion of the estimated (posterior median) overall global population mainly using polluting fuels.

the Sub-Saharan Africa region, where use of polluting fuels as the main fuel for cooking has only dropped from 90% [87–92] in 1990 to 84% [82–86] in 2020. If observed trends continue, this is projected to only drop to 81% [76–85] in 2030, meaning four in five Sub-Saharan African people will continue to suffer the health and socioeconomic burdens of polluting cooking (this figure would likely be higher if it accounted for stove stacking).

Once again, to truly understand the human cost of polluting cooking, it is more telling to consider the absolute number of people mainly using polluting fuels (Fig. 4). The number of people mainly cooking with polluting fuels is rising at an alarming rate in Sub-Saharan Africa and is projected to exceed 1 billion people by as soon as 2025.

In the year 2000, out of those mainly cooking with polluting fuels, 3 in 4 (75%) lived in either Central Asia and Southern Asia or Eastern Asia and South-eastern Asia, and only 1 in 5 (19%) resided in Sub-Saharan Africa, as illustrated in Fig. 5. In 2020, around 1 in 3 (34%) lived in Sub-Saharan Africa and this is projected to approach to 1 in 2 (44%) by 2030.

**The changing fuel mix in low-income and middle-income countries**. Analysis of specific fuel use at regional, country, and sub-national levels can help to better estimate the impacts of current policies for household energy use as well as inform the future development of policies and programs. Here, we discuss some of the most notable trends across LMICs.

Among LMICs (Fig. 6), use of gaseous fuels as the main cooking fuel increased consistently from 31% [23–41] in 1990 to 49% [41–56] in 2020, overtaking unprocessed biomass fuels as the dominant main cooking fuel type in the last decade. Use of

electricity as the main cooking fuel also rose, from 4% [3–7] in 1990 to 8% [4–14] in 2020, with a considerably larger increase in urban areas where infrastructure tends to be better established.

Between 1990 and 2010, increases in the use of clean fuels as the main cooking fuel appear to be principally explained by considerable decreases in the use of coal and kerosene as the main fuel. Use of coal as the main fuel in rural areas has dropped from 12% [3–25] in 1990 to 5% [3–8] in 2010 then to 2% [1–6] in 2020. Use of kerosene as the main fuel has also decreased: in urban areas it dropped from 10% [8–12] in 1990 to 4% [3–5] in 2010 then to 2% [1–3] in 2020, while in rural areas it dropped from 3% [2–5] in 1990 to 1% [0–2] in 2020. However, from around 2010 onwards use of biomass as the main fuel has also started to decrease consistently, primarily in rural areas where use of unprocessed biomass as the main fuel has dropped from 68% [63–73] in 2010 to 60% [51–68] in 2020.

Although globally use of kerosene as the main fuel has dwindled, it persists in urban areas of LMICs in both Oceania (15% [7–35] in 2018) and in Sub-Saharan Africa (6% [4–9] in 2020). Globally the proportion mainly using charcoal is low (4% [3, 4] in 2020), but in urban areas of Sub-Saharan Africa (Fig. 7) it has overtaken biomass as the most popular main fuel (30% [25–35] in 2020). If observed trends continue into the next decade, in urban areas of LMICs use of gaseous fuels as the main fuel is projected to start falling as more people switch to electricity as their main fuel, and eventually level-off overall.

**Case studies: regional and country analyses of fuel use.** Here, we demonstrate how our estimates can be used for detailed analysis at the regional, national, and sub-national level, using

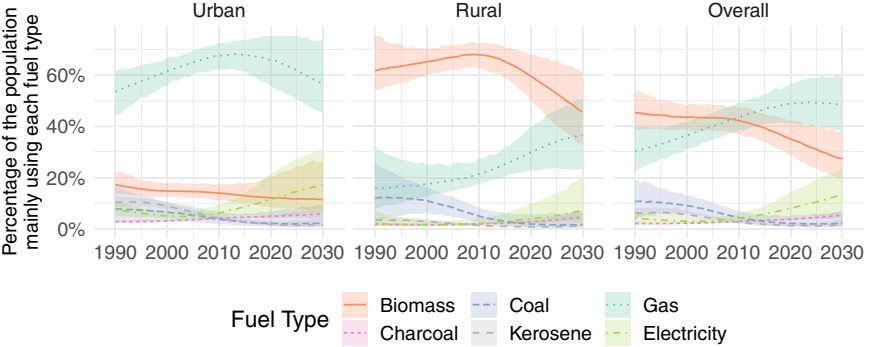

**Fig. 6 Cooking fuel use in LMICs.** Estimated (posterior median) percentage of the population in LMICs mainly using each fuel type, with 95% uncertainty intervals.

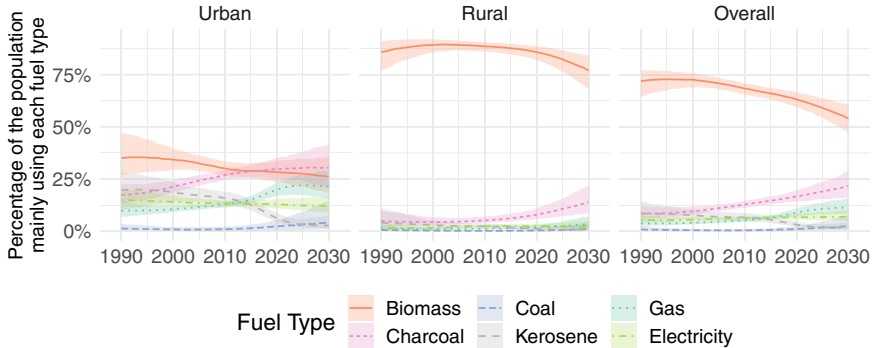

**Fig. 7 Cooking fuel use in Sub-Saharan Africa.** Estimated (posterior median) percentage of the population in Sub-Saharan Africa mainly using each fuel type (lines), with 95% uncertainty intervals (shaded areas). Plots for other regions are included in the Supplementary Information.

Sub-Saharan Africa and Ghana as case studies. In 2020, main fuel use in urban areas of Sub-Saharan Africa (Fig. 7) is highly pluralistic, consisting of charcoal (30% [25–35]), biomass (28% [24–34]), gaseous fuels (20% [17–23]), and electricity (13% [11–15]). In rural areas, however, use of biomass (86% [82–89]) and charcoal (8% [6–11]) as the main fuel constitutes a near duopoly, with only 6% using any of the other fuels as the main fuel.

If observed trends continue into the next decade, use of kerosene as the main fuel is projected to diminish to around 2% of the Sub-Saharan African population in 2030, with only a few countries maintaining high levels in 2030: 36% in Equatorial Guinea, 44% in Djibouti, and 76% in Sao Tome and Principe. In fact, in Sao Tome and Principe use of kerosene as the main fuel is projected to increase. Meanwhile, modest decreases in the use of biomass as the main fuel are likely to be largely offset by increases in the use of charcoal as the main fuel. Concerningly, very little progress is projected to be made in the use of gaseous fuels or electricity as the main cooking fuel either in urban or rural parts of Sub-Saharan Africa.

Zooming in to the national and sub-national level, Fig. 8 shows modeled estimates for main fuel use in Ghana alongside observed values from available household survey data—these are plotted to illustrate how the model captures non-linear fuel use trends, survey variability, and associated uncertainty.

In Ghana, the plurality of people mainly used biomass fuels in 2020 (38% [25–52]), with a further 30% [19–43] mainly relying on charcoal (Fig. 8). Use of biomass as the main fuel remains high in rural areas, despite dropping from 90% [80–97] in 1990 to 68% [51–82] in 2020. Although main use of charcoal was steadily rising in rural areas between 1990 (8% [2–18]) and 2010 (17% [10–26]), there is some evidence that this has stalled. In urban areas, meanwhile, main use of gaseous fuels has risen consistently

from 5% [2–10] in 1990 to 44% [28–61] in 2020. This is likely the result of concerted government efforts (starting around 1990) to promote the use of LPG as a substitute for the widely used charcoal and firewood[14]. Increased use of gas as the main cooking fuel has come at the expense of biomass, which dropped about 14% points between 1990 and 2010, and charcoal, which dropped about 15% points between 2010 and 2020. Indeed, there is some evidence (65% probability) that in 2020 more people mainly used gaseous fuels than any other fuel in urban areas of Ghana. If observed trends continue, main use of gaseous fuels is projected to rise to 46% by 2030, meaning about 1 in 2 people in Ghana will still rely mainly on polluting fuels for cooking.

## Discussion

Previous estimates of clean versus polluting/solid fuel use for cooking have played a vital role in informing global efforts to address the global energy injustice of household air pollution. However, by combining increasingly detailed survey data with advanced statistical modeling approaches, we have produced new estimates based on specific fuels. These estimates offer more detailed assessment of progress towards global goals and work to maximize the utility of data capturing household energy use and its impacts on health for policymaking. In particular, a greater understanding of what fuels people are using specifically can help pre-empt barriers to future adoption of clean cooking (e.g., affordability constraints or cultural preferences). Here, we used a novel Bayesian hierarchical modeling approach to comprehensively and reliably estimate the use of six fuel types (as the main cooking fuel)—as well as overall clean and polluting fuel use—under realistic and plausible constraints, from 1990 to 2020. We also presented future projections of existing trends up to 2030, representing a "business-as-usual" scenario to motivate new

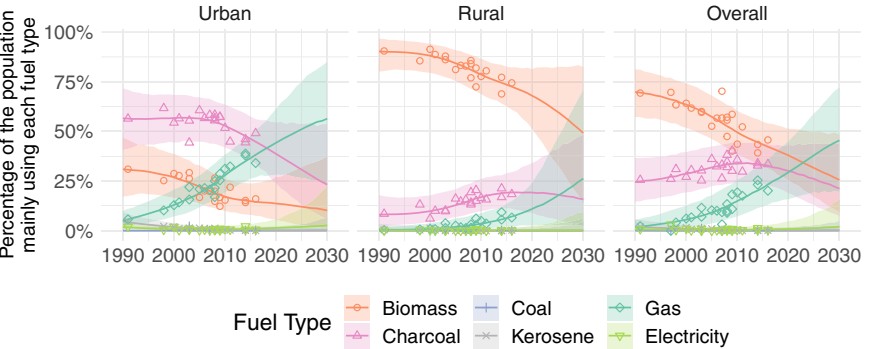

**Fig. 8 Cooking fuel use in Ghana.** Estimated (posterior median) percentage of the urban population (left), the rural population (center) and overall population (right) of Ghana mainly using each fuel type, with central estimates as lines. Points show available survey data. The 95% uncertainty intervals shown as shaded areas combine model uncertainty and survey variability: where data are plentiful, the uncertainty is small and the intervals capture the vast majority of survey points, where survey data are limited or unavailable, in particular when projecting into the future, the uncertainty grows, and our uncertainty intervals are wider. Plots for other LMICs are included in the Supplementary Information.

policy and providing a baseline against which the effects of new interventions can be assessed.

Our analysis shows that, although there has been progress towards clean household energy, the global community is far off track from reaching universal access to clean cooking by 2030. The global proportion mainly using polluting fuels dropped by an estimated 17% between 1990 and 2020, although the absolute number of people using polluting fuels has deviated little from 3 billion over the last three decades. Global progress in urban areas is now static, although use of clean fuels as the main cooking fuel is increasing in rural areas (particularly gas and electricity). Indeed, our business-as-usual scenario projects that 2.7 billion people—just under 1 in 3—will continue to mainly rely on polluting cooking fuels in 2030. A deeper regional analysis has highlighted the emergence of Sub-Saharan Africa as having the largest population mainly using polluting fuels for cooking after 2020, which is likely to exceed 1 billion people by 2025 under a business-as-usual scenario; the need for greater focus and resources to implement policies and programs promoting the adoption of clean cooking in Sub-Saharan Africa cannot be overstated. Analysis at the level of specific fuels reveals further insights, such as the global elimination of kerosene and coal for cooking, and the emergence of charcoal as the most popular fuel in urban Sub-Saharan Africa.

While the availability of a complete set of estimates by specific fuel represents a significant step forward in the monitoring and understanding of polluting fuel use for cooking, these estimates do not take into account the technology used for cooking nor supplementary cooking fuels and technologies—due to a lack of data from nationally-representative surveys. Moving forward, access to technological solutions like low-emission advanced combustion biomass cookstoves should be monitored in national surveys to facilitate inclusion in global analyses. These new surveys should follow the example of the Core Questions on Household Energy Use, jointly developed by the WHO and the World Bank's Energy Sector Management Assistance Program (ESMAP) to track SDG Target 7.1[15]. Our estimates also do not currently account for stove-stacking, where households use more fuels than just the main cooking fuel. This is an important issue, noting for instance that a household using clean fuels 51% of the time will still suffer significant negative health and social impacts from using polluting fuels 49% of the time, despite being counted as "mainly using clean fuels". Quantifying the health impacts of stove-stacking will rely on: enhanced and harmonized data collection capturing the fuels and technologies used in the home for all major end-uses including cooking, heating, and lighting; and

robust epidemiological evidence quantifying health risk from specific fuels and technologies used. Enhanced monitoring efforts paired with future modeling that accounts for stove-stacking will improve understanding of exposure to total household air pollution, thus better informing policy and programmatic decision-making, as well as the global monitoring of health and environmental impacts.

## Methods

**Household survey data and selection criteria.** Data used in this analysis are drawn from the WHO's Household Energy Database[16], a regularly updated compilation of nationally-representative household survey data for WHO Member States from various sources, detailed in Supplementary Table 1. Surveys in the database were downloaded manually and collated using Microsoft Excel (version 16.50) and occasionally Stata/SE (version 15.1). The version of the database used for this analysis (30th January 2020) comprises 1353 surveys collected from a total of 170 countries (including high income countries) between 1960 and 2018. For this analysis we exclude surveys from before 1990, and only include data from surveys providing individual fuel breakdowns and with less than 15% of the population in total categorized as "missing", "not cooking in the household", or "mainly cooking with "other fuels". There was no differentiation in the model between surveys that reported only household-weighted or population-weighted fuel use estimates. Where surveys reported both household-weighted and population-weighted estimates, only population-weighted estimates were used, in order to best estimate the population reliant on different cooking fuels. Using this selection criteria, 1136 surveys—collected from 153 countries—were used for modeling. Supplementary Table 1 shows both the number of surveys in the database and the number used for modeling from each data source. Meanwhile, Supplementary Table 2 shows the number of survey data points excluded for failing to meet inclusion criteria.

Surveys included in the database are inconsistent in the questions posed to households about cooking (typical questions by survey source are included in Supplementary Table 1). Most survey questions focus on the main or primary type of cooking fuel or energy rather than the cooking device, and thus the database version included in this study does not contain comprehensive data on solid fuel stove type (e.g., forced draft, brand information). Almost all surveys only assess the primary, or main, cooking fuel, or energy source which constrains the analysis to the primary fuel and technology used for cooking, although it is well documented that households often "stove-stack" or use multiple stoves and/or fuels[17–19]. Most surveys report the percentage of respondents mainly using each fuel separately for urban and rural areas. The definitions of urban and rural may vary by country, and we adopt these reported values directly rather than applying any standard definition of urban and rural.

The WHO Household Energy Database contains data on the proportion of households mainly using a wide variety of cooking fuels, including alcohol fuels (e.g., ethanol), biogas, charcoal, coal, crop residues, dung, electricity, kerosene, liquid petroleum gas (LPG), natural gas, solar energy, and wood. However, surveys are not always consistent in the fuel options they present to respondents. In particular, some surveys combine fuels into a single option (notably natural gas and LPG are often combined into the category "gas"). The result of this is that the time series of survey data for certain individual fuels can be unstable or unreliable in some countries.

Where appropriate in terms of similarity of health impacts, and relevance to policymakers, these issues can be remedied by combining affected fuels into a

single category for modeling purposes. Here, we combine wood, crop residues, and dung into the category "biomass", representing the combined use of unprocessed/raw biomass fuels, and we combine LPG, natural gas and biogas into the category "gas"—refer to Fig. 1 for a visual representation of these categories. Although solar and ethanol are considered clean fuels, they have been included under the category "other fuels", due to the sparse number of data points available for these fuels (105 total data points for solar energy ranging between 0 and 0.8%; seven total data points for ethanol ranging between 0 and 0.14%).

We therefore estimate the population mainly using six fuel types: 1. biomass, 2. charcoal, 3. coal, 4. kerosene, 5. gas, and 6. electricity. A final category, "other fuels" represents the aggregate use of minor clean fuel types, e.g., solar and ethanol. Estimates for overall "polluting" and overall "clean" fuel use are then derived by aggregating estimates of relevant fuel types. "Other fuels" were not modeled individually but are included in the aggregate "clean" category.

**The global household energy model.** Previous statistical models for estimating fuel use have focussed on a single variable, i.e., solid fuel use or polluting fuel use[9,20]. Instead, we sought to model how a strongly related set of variables (the proportion of the population using each individual fuel type) changes over time, under the key constraint that as the use of one fuel increases the sum of the others must decrease, so that the total never exceeds 100%. No standard statistical procedure is available to achieve this while also properly quantifying the uncertainty associated with estimates for each fuel, which merited the development of the bespoke Global Household Energy Model[11] (GHEM), a state-of-the-art Bayesian hierarchical approach[21] to jointly estimating the use of individual fuels for cooking.

Trends in the proportions using each fuel type are modeled together for both urban and rural areas of each country using smooth functions of time (thin-plate splines) as the only covariate. Estimates produced by the model are realistic in the sense that, for each country, urban, rural, and overall fuel use is linked by estimates of the survey sample urban proportion (including for years without surveys), also based on smooth functions of time.

The model outputs Bayesian "posterior" probability distributions for fuel use in a given year and country, which can be used to answer questions like "What is the probability that the use of coal exceeds 10% in urban areas of Mongolia?". For reporting purposes, summaries of these distributions can be taken to provide both point estimates (e.g., means or medians, the latter being what we present here and in the Supplementary Information/Data) and measures of uncertainty (e.g., 95% prediction intervals (PIs))—which mean there is a 95% probability that fuel use lies within the given range). Here, we use the term "uncertainty interval" to describe central 95% posterior credible/prediction intervals.

GHEM is implemented using custom code (fully provided in Supplementary Software 1) in the R programming language (version 4.0.0) and the NIMBLE[22] software package (version 0.10.1) for Bayesian statistical modeling with Markov chain Monte Carlo (MCMC). We also used the following R packages for our analysis: abind (1.4-5); coda (0.19-3); doParallel (1.0.15); ggfan (0.1.3); ggplot2 (3.3.0); grid (4.0.0); gridExtra (2.3); mgcv (1.8-33); openxlsx (4.1.5); Rcolorbrewer (1.1-2); readxl (1.3.1); reshape2 (1.4.4); rgdal (1.5-16); scales (1.1.0); and tidyverse (1.3.0). The version of GHEM used for this analysis differs from the previously published version[11] in that no regional structures were assumed a-priori. Non-informative prior distributions were assumed for all model parameters[11]. We ran four MCMC chains from distinct randomly generated sets of initial values, using different random number generator seeds for each chain. We ran the chains for 80,000 iterations, discarding the first 40,000 from each chain as "burn-in" and then thinning by a factor of 40 to reduce system memory usage. The result is a total of 4000 posterior samples for each model parameter, which are used to calculate posterior medians and central 95% posterior credible/prediction intervals.

The probability distributions assumed for input survey data do not allow for inputs where the sum of the percentage mainly using all mutually exclusive fuel categories exceeds 100% (110 surveys, with a median total excess of 0.01%), which can occur due to rounding at different stages of data collection. For these surveys, fuel use values were uniformly scaled (divided by the sum of mutually exclusive categories), to have a total of 100%. Countries classified as high-income according to the World Bank country classification[23] (60 countries) are assumed to have fully transitioned to clean household energy and are reported as >95% access to clean fuels and technologies[1]. In addition, no estimates are provided for LMICs where no surveys were available or suitable for modeling post-1990 (Bulgaria, Cuba, Lebanon, and Libya). Modeled estimates for the use of overall clean, overall polluting and specific fuels are therefore provided for a total of 130 countries—128 LMICs plus two countries with no World Bank income classification (Cook Islands and Niue).

Population data from the United Nations Population Division (2019 version) were used to derive the population-weighted regional and global aggregates. We present aggregate estimates for the eight SDG regions, as well as for the six WHO regions. LMICs without suitable survey data were excluded from all regional calculations and high-income countries were excluded from regional calculations for specific fuels—this means our regional estimates for specific fuels (e.g., gas) refer only to LMICs in those regions. Values of 100% clean fuel use were used for high income countries when calculating regional aggregates of clean and polluting fuel use.

**Future projections.** We also project observed trends in fuel use into the future using GHEM. These future projections were developed by extrapolating observed trends, representing a "business-as-usual" scenario assuming no new policies or interventions.

The degree of uncertainty associated with such projections depends on a number of factors which vary by country, including the number of surveys conducted near present day and how changeable the trends are estimated to be over the available data period (1990–2018)—for example, projections for a country where trends are linear may display less uncertainty than a country with sudden changes in fuel use (e.g., Indonesia). The model has been validated[11] for making fuel use predictions up to 5 years beyond the end year of the data. Hence for years close to the end of the data period (e.g., 2019, 2020, 2021), point estimates and 95% prediction intervals can be interpreted as predictions of what may happen based on trends in the data. Further into the future, uncertainty tends to grow beyond practical levels but point estimates remain useful for policy purposes with a specific interpretation: what may happen if observed trends continue and no new policies or interventions are introduced.

**Health impacts.** Our estimates of the populations mainly using polluting fuels for cooking are used by the WHO to estimate the global burden of disease from household air pollution[3]. Future WHO burden of disease estimates are anticipated to be calculated based on estimated populations mainly using specific fuels and technologies for cooking.

Other institutions have also developed burden of disease estimates for household air pollution based on cooking fuels, all with varying results but ultimately telling the same message: millions of premature deaths annually and hundreds of millions of years of healthy life lost due to exposure to household air pollution[24–26].

**Disclaimer.** The authors alone are responsible for the views expressed in this article and they do not necessarily represent the views, decisions or policies of the institutions with which they are affiliated.

**Reporting summary.** Further information on research design is available in the Nature Research Reporting Summary linked to this article.

## Data availability

To reproduce the results in this study, the relevant version of the WHO Household Energy database from the 30th of January 2020 is available from the corresponding author or directly from the WHO (householdenergy@who.int) on reasonable request, on the condition that the correspondence states what the data set will be used for. The data generated in this study are provided as Supplementary Data. Future updates to household cooking fuel estimates will be posted to the WHO Global Health Observatory data repository (https://www.who.int/data/gho/data/themes/air-pollution/household-air-pollution).

## Code availability

Custom R code (tested using R version 4.0.0) to reproduce the results in this study is provided for download as Supplementary Software 1. Running the code may require up to 64 gigabytes of system memory. Please note the analysis relies on Markov Chain Monte Carlo and Monte Carlo simulation methods, which are both stochastic in nature. This means that figures and quoted statistics have the potential to differ slightly each time the code is executed.

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

## Acknowledgements

The authors gratefully acknowledge funding from an ESRC Impact Accelerator Award [ES/T501906/1] (Sub-Award) Knowledge Exchange Fellowship awarded to Oliver Stoner. The financial support of ESMAP is also gratefully acknowledged. ESMAP is a partnership between the World Bank and partners to help low- and middle-income countries reduce poverty and boost growth through sustainable energy solutions. Theo Economou has received funding from the European Union's Horizon 2020 research and innovation programme under grant agreement No. 856612 and the Cyprus Government. Finally, the authors thank Gavin Shaddick for originally establishing the collaboration with the WHO which enabled this work.

## Author contributions

Oliver Stoner, Jessica Lewis, Itzel Lucio Martínez, and Heather Adair-Rohani developed the concept for the manuscript. Oliver Stoner developed the Global Household Energy Model, carried out the country and regional analysis and made major contributions to the manuscript. Jessica Lewis made significant contributions to the manuscript. Itzel Lucio Martínez was primarily responsible for updating the WHO Household Energy Database—including the addition of new surveys and validation of existing surveys—during the period over which this work was carried out and made significant contributions to the manuscript. Sophie Gumy contributed to the concept of the work, previously managed the WHO Household Energy Database and made minor contributions to the manuscript. Theo Economou previously supervised Oliver Stoner in developing the model and made minor contributions to the manuscript. Heather Adair-Rohani made significant contributions to the manuscript, manages the WHO Household Energy Database, and coordinated efforts for the development and application of the presented methodology for global household energy monitoring.

## Competing interests

The authors declare no competing interests.
