## [Peer Review File · Nature Communications]

Household cooking fuel estimates at global and country level for 1990 to 2030REVIEWERS' COMMENTS

Reviewer #1 (Remarks to the Author):

Overall, I found this paper quite useful and well done. The new estimates are useful and the modeling is generally done well. However, I do have some suggestions for improvement:

The primary weakness of this paper is the lack of a robust discussion of how people use cooking fuels. It was only in the supporting information that the authors clearly stated what they mean by "use" of a certain cooking fuel. This lack of clarity can make the paper potentially misleading, and it needs to be improved before publication.

Already in the abstract I would like to see a clear statement of what is measured. The answer seems to be the fuel that the household respondent – who may or may not be the primary cook? – considers "primary".

Another comment on abstract: it's really hard to understand statements like "lack access to clean cooking in 2030" unless there is a definition. Does this mean they never use clean cooking? Or does this mean it's not their primary fuel?

In general, I find this paper to have quite a bit of imprecise language. Here is another example: "If observed trends continue into the next decade, use of kerosene is projected to diminish to around 2% of the Sub-Saharan African population in 2030, with only a few countries maintaining high levels in 2030: 36% in Equatorial Guinea, 44% in Djibouti, and 76% in Sao Tome and Principe – where kerosene use is actually projected to increase." Here writing seems to suggest households stop using the fuel, but that doesn't seem to be what the surveys capture. So is this "primary use" or "any use"?

I would like to see somewhere a more prominent statement on the essential importance of exclusive clean cooking. The health impacts from clean cooking are nonlinear, and if "mainly using" means 51% then the health benefits will be limited.

In Section 1, this issue needs to be discussed in some depth. Before getting to numbers, include 1-2 paragraphs on what is at stake here. I suggest to include the following:

Fuel stacking is very common. Households use solid fuels because they are often the most available, affordable, and convenient choices. Consider citation: Shankar, Anita V., Ashlenn K. Quinn, Katherine L. Dickinson, Kendra N. Williams, Omar Masera, Dana Charron, Darby Jack et al. "Everybody stacks: Lessons from household energy case studies to inform design principles for clean energy transitions." *Energy Policy* 141 (2020): 111468.

Households vary cooking fuels depending on dish, season, access, etc. This is a very complex problem, and any idea of a "primary cooking fuel" is a major simplification. Consider citation: Gould, Carlos F., and Johannes Urpelainen. "LPG as a clean cooking fuel: Adoption, use, and impact in rural India." *Energy Policy* 122 (2018): 395-408.

In the discussion section, this lead sentence is underwhelming:

"Previous estimates of cooking fuel use have been limited to clean versus polluting/solid fuel use, preventing detailed assessment of progress towards global goals and minimizing the utility of household energy use data for policymaking."

I don't quite understand why we need to know about 6 different fuels to assess progress toward global goals. I also don't understand why this would "minimize" utility. A little bit of precision, humility, and recognition of earlier work's value would help here.

Discuss the challenge of the WHO surveys in greater depth. A short note on this somewhere in the paper, then extensive discussion in SI. Remember the golden rule of modeling: garbage in, garbage out. The modeling improvements in this paper are nice, but the primary uncertainties come from the extreme difficulty of surveying households on this. I don't see this as an obstacle to publication, but the authors need to be clear here and not leave a false sense of precision.

Reviewer #2 (Remarks to the Author):

This is an important, thorough, and interesting paper. Like previous modeling exercises led by WHO, it will play an important role in shaping the narrative around polluting fuel use for years to come. I have a number of mainly small comments and suggestions.

- Define rural and urban somewhere, or point to WHO definitions of rural and urban, if such things exist.
- It seems important to acknowledge the existence of other BoD exercises, such as those run by IHME. What fuel use estimates are used there, and how do they relate to these?
- It would be beneficial to point to/reference the code provided in Stoner et al 2020 and provide code for replication here, if possible.
- Given the inclusion of much of the survey data you cite in Table 1, could you include a second, short table on with a little more detail on why some survey data was excluded?
- The point about mixed stove use - and its implications for these estimates and more broadly - should be made more clearly and made more pronounced

Minor.

Abstract - line 16: can you specific whether this is any reliance or exclusive reliance?

L32 - Can you cite something for the hours per week claim, risk of MSK injury, violence?

L60 - put quotation marks around "polluting and clean fuel use" — otherwise this sentence is hard to follow

Figure 1: Great! Why the empty circle at the top? LPG is a gas when used as a fuel, but a liquid under pressure / in the canister. Not that I need to tell you this! If you're going to include an empty circle under clean other fuels, you may consider doing the same for solid fuels — for instance, plastic/waste burning.

L132 - change level to levels

L169 - Kerosene use is projected to increase in ST and Principe? Or the entire list of countries? Any explanation?

L263 - Data are...

Response to reviewers' comments

Reviewer #1 (Remarks to the Author):

Overall, I found this paper quite useful and well done. The new estimates are useful and the modeling is generally done well. However, I do have some suggestions for improvement:

1. The primary weakness of this paper is the lack of a robust discussion of how people use cooking fuels. It was only in the supporting information that the authors clearly stated what they mean by “use” of a certain cooking fuel. This lack of clarity can make the paper potentially misleading, and it needs to be improved before publication.

We have added a paragraph to the introduction where we discuss the issue of fuel and stove stacking and the limitations of estimates which only capture which fuel a household mainly uses.

2. Already in the abstract I would like to see a clear statement of what is measured. The answer seems to be the fuel that the household respondent – who may or may not be the primary cook? – considers “primary”.

We have modified the abstract to make it clear that we are providing estimates of the fuels mainly used by households.

3. Another comment on abstract: it’s really hard to understand statements like “lack access to clean cooking in 2030” unless there is a definition. Does this mean they never use clean cooking? Or does this mean it’s not their primary fuel?

This sentence has been modified to clarify that we are talking about the population that will still mainly use polluting fuels for cooking.

4. In general, I find this paper to have quite a bit of imprecise language. Here is another example: “If observed trends continue into the next decade, use of kerosene is projected to diminish to around 2% of the Sub-Saharan African population in 2030, with only a few countries maintaining high levels in 2030: 36% in Equatorial Guinea, 44% in Djibouti, and 76% in Sao Tome and Principe – where kerosene use is actually projected to increase.” Here writing seems to suggest households stop using the fuel, but that doesn’t seem to be what the surveys capture. So is this “primary use” or “any use”?

We have made affected sentences more precise by using phrases such as “mainly using” or “as the main fuel for cooking”.

5. I would like to see somewhere a more prominent statement on the essential importance of exclusive clean cooking. The health impacts from clean cooking are nonlinear, and if “mainly using” means 51% then the health benefits will be limited.

We have added a sentence to this effect in the Discussion.

6. In Section 1, this issue needs to be discussed in some depth. Before getting to numbers, include 1-2 paragraphs on what is at stake here. I suggest to include the following:

Fuel stacking is very common. Households use solid fuels because they are often the most available, affordable, and convenient choices. Consider citation:

Shankar, Anita V., Ashlinn K. Quinn, Katherine L. Dickinson, Kendra N. Williams, Omar Masera, Dana Charron, Darby Jack et al. "Everybody stacks: Lessons from household energy case studies to inform design principles for clean energy transitions." *Energy Policy* 141 (2020): 111468.

Households vary cooking fuels depending on dish, season, access, etc. This is a very complex problem, and any idea of a "primary cooking fuel" is a major simplification. Consider citation:

Gould, Carlos F., and Johannes Urpelainen. "LPG as a clean cooking fuel: Adoption, use, and impact in rural India." *Energy Policy* 122 (2018): 395-408.

We have added a paragraph to the introduction which states very clearly the significance of stove stacking and the limitations of estimates based only on which fuels households are mainly using. We thank the reviewer for the suggested references, which we have included. We have also some more comments on stove stacking in the Discussion.

7. In the discussion section, this lead sentence is underwhelming:

"Previous estimates of cooking fuel use have been limited to clean versus polluting/solid fuel use, preventing detailed assessment of progress towards global goals and minimizing the utility of household energy use data for policymaking."

I don't quite understand why we need to know about 6 different fuels to assess progress toward global goals. I also don't understand why this would "minimize" utility. A little bit of precision, humility, and recognition of earlier work's value would help here.

Please see the new text at the beginning of the discussion, which imparts greater recognition of the value of previous estimates and expands on why fuel-specific estimates can help to better inform policymaking.

8. Discuss the challenge of the WHO surveys in greater depth. A short note on this somewhere in the paper, then extensive discussion in SI. Remember the golden rule of modeling: garbage in, garbage out. The modeling improvements in this paper are nice, but the primary uncertainties come from the extreme difficulty of surveying households on this. I don't see this as an obstacle to publication, but the authors need to be clear here and not leave a false sense of precision.

Reviewer #2 (Remarks to the Author):

This is an important, thorough, and interesting paper. Like previous modeling exercises led by WHO, it will play an important role in shaping the narrative around polluting fuel use for years to come. I have a number of mainly small comments and suggestions.

9. Define rural and urban somewhere, or point to WHO definitions of rural and urban, if such things exist.

We have clarified in Section 4.1 that the definition of urban and rural can vary by country, and we directly adopt the urban and rural values reported by surveys, rather than applying any standard definition.

11. It seems important to acknowledge the existence of other BoD exercises, such as those run by IHME. What fuel use estimates are used there, and how do they relate to these?

The work presented here is not a burden of disease exercise, however our estimates can be and are used as inputs for BoD calculation. We have added a short subsection (4.4) detailing this, noting other efforts e.g. IHME which estimate fuel use with largely the same survey data sources.

12. It would be beneficial to point to/reference the code provided in Stoner et al 2020 and provide code for replication here, if possible.

We have now provided the code used for this analysis in a ZIP file uploaded alongside the manuscript.

13. Given the inclusion of much of the survey data you cite in Table 1, could you include a second, short table on with a little more detail on why some survey data was excluded?

We have prepared a new table (Table 2) detailing the number of survey data rows excluded for each specific reason.

14. The point about mixed stove use - and its implications for these estimates and more broadly - should be made more clearly and made more pronounced

We have addressed this point in our response to comments 1 and 6 by Reviewer #1.

15. Abstract - line 16: can you specific whether this is any reliance or exclusive reliance?

We have clarified that they “mainly used” polluting fuels.

16. L32 - Can you cite something for the hours per week claim, risk of MSK injury, violence?

We have now cited the 2016 WHO report *Burning opportunity: clean household energy for health, sustainable development, and wellbeing of women and children*.

17. L60 - put quotation marks around “polluting and clean fuel use” — otherwise this sentence is hard to follow

Done.

18. Figure 1: Great! Why the empty circle at the top? LPG is a gas when used as a fuel, but a liquid under pressure / in the canister. Not that I need to tell you this! If you’re going to include an empty circle under clean other fuels, you may consider doing the same for solid fuels — for instance, plastic/waste burning.

The circle at the top is just an artistic element which we don’t believe distracts from the information in the plot. We have removed the empty circle under other fuels, as we think this does distract from the fuels which are within the scope of the model.

19. L132 - change level to levels

Done.

20. L169 - Kerosene use is projected to increase in ST and Principe? Or the entire list of countries? Any explanation?

We have clarified that it is projected to increase in Sao Tome and Principe.

21. L263 - Data are...

Changed.